# A Quantitative Study of Micro and Macro Mechanical Parameters Based on the PFC^3D^ Flat-Joint Model

**DOI:** 10.3390/ma15196790

**Published:** 2022-09-30

**Authors:** You-Liang Chen, Yun-Gui Pan, Xi Du, Qi-Jian Chen, Shao-Ming Liao, Ning Zhang, Su-Ran Wang, Bin Peng

**Affiliations:** 1Department of Civil Engineering, School of Environment and Architecture, University of Shanghai for Science and Technology, No. 516, Jungong Road, Yangpu District, Shanghai 200093, China; 2Department of Geotechnical Engineering, Tongji University, 1239 Siping Road, Shanghai 200092, China

**Keywords:** PFC^3D^, rock, flat-joint model, orthogonal experimental design, micro and macro mechanical parameters calibration, flexible boundaries

## Abstract

The flat-joint model, which constructs round particles as polygons, can suppress rotation after breakage between particles and simulate more larger compression and tension ratios than the linear parallel-bond model. The flat-joint contact model was chosen for this study to calibrate the rock for 3D experiments. In the unit experiments, the triaxial unit was loaded with flexible boundaries, and the influence of each microscopic parameter on the significance magnitude of the macroscopic parameters (modulus of elasticity *E*, Poisson’s ratio *ν*, uniaxial compressive strength *UCS*, crack initiation strength *σ*_ci_, internal friction angle *φ* and uniaxial tensile strength *TS*) was analysed by ANOVA (Analysis of Variance) in an orthogonal experimental design. Among them, *E*_ƒ_, *k*_ƒ_ has a significant effect on *E*; *C*_ƒ_ and *k*_ƒ_ have a significant effect on *ν*; *C*_ƒ_, *σ*_ƒ_ and *k*_ƒ_ have a significant effect on *UCS*; *C*_ƒ_; *σ*_ƒ_ and *E*_ƒ_ have a significant effect on TS; *R*_sd_ has a significant effect on *σ*_ci_; and *φ*_f_, *E*_ƒ_, *k*_ƒ_, *μ*_ƒ_, and *σ*_ƒ_ have a significant effect on *φ*. Regressions were then carried out to establish the equations for calculating the macroscopic parameters of the rock material so that the three-dimensional microscopic parameters of the PFC can be quantitatively analysed and calculated. The correctness of the establishment of the macroscopic equations was verified by comparing the numerical and damage patterns of uniaxial compression, Brazilian splitting, and triaxial experiments with those of numerical simulation units in the chamber.

## 1. Introduction

With the development of science and technology, people are exploring underground space more and more deeply. Using computers to analyse the mechanical properties of discontinuous media underground has become a major research tool. Liu et al. [1] used deep recurrent neural nets and convolutional neural networks for vibration-based working face ground recognition. Deep natural rocks are subjected to three-dimensional stresses and the presence of fractures and joints in the rocks is potentially harmful under external loads. These problems are studied by means of computer modeling techniques to analyze microscopic fractures in rocks [2,3], and the methods commonly used today include DEM, FEM-DEM, lattice, embedded discontinuities and granular flow. The following authors have conducted relevant studies using the above approaches, such as Nikolić et al. [4], who treated materials as disordered, inhomogeneous, and multiphase through a lattice model that simulates damage phenomena in quasi-brittle materials (e.g., concrete or rock) at fine or microscopic scales. Mahabadi et al. [5] validated a Brazilian splitting experimental microdimensional model based on a combined finite discrete element method (FDEM) with a new hybrid FDEM code that accurately estimates crack trajectories and damage mechanisms of specimens and simulates cliff recession as well as complex damage mechanisms of failed cliffs. Nikolic et al. [6] applied the embedded discontinuity beam lattice model, which treats rocks as two-phase composites, where intact rocks and rocks with pre-existing microcracks and other defects were used to simulate the propagation of cracks in rocks. Leandro et al. [7] further extended the RBSN formulation based on the rigid-body spring network method and used the model to perform numerical direct tensile tests, Brazilian splitting disc tests, triaxial tests, and lateral limit-free compression tests, and the results showed that the model could better match the macroscopic complex damage phenomena. In this paper, PFC^3D^ is used to analyze the relationship between macroscopic parameters, rocks considered as a collection of discrete granular bodies, and the distribution of cracks on the rock microscale. The damage patterns are explored through the fracture of microscopic particle contact bonds to produce cracks.

Chong et al. [8] carried out a fine-scale simulation of marble deformation and the damage process under different stress paths by PFC^3D^. Based on the relationship between the three types of displacement fields and the fracture surface during crack formation, a fine-scale fracture surface fitting method based on numerical simulation was proposed to extract and reconstruct the final fracture surface of the specimen. At the beginning of the micro-parameter studies of PFC, most scholars [9,10,11] used parallel-bonding models (PBM) to simulate rocks and analysed the influence of the micro-parameters of each parallel-bonding model on the macro-parameters. Liu et al. [12] established uniaxial and biaxial numerical simulations of rock materials by PFC^2D^ and derived a linear relationship between the parallel bond modulus and the Young’s modulus and the normal/shear stiffness ratio to Poisson’s ratio of the simulated materials. It is pointed out that the compressive strength of the material is mainly affected by the normal strength when the ratio of the normal strength to the tangential strength of the parallel bond is greater than two, and the compressive strength of the material is mainly affected by the tangential strength when the ratio of the normal strength to the tangential strength is less than two. Erdi et al. [13] investigated the correlation of macroscopic mechanical parameters of parallel cohesive models. Zhao et al. [14] concluded that the macroscopic elastic modulus of the model was mainly determined by the particle contact of Young’s modulus and particle cohesion of Young’s modulus, and Poisson’s ratio was mainly determined by the particle stiffness, which is logarithmically related and less influenced by the particle size through PFC^2D^ numerical simulation. The compressive strength is mainly determined by the ratio of the normal to tangential bonding stress of the particles. Liu et al. [15] investigated the effects of the friction and rotation coefficients on the natural resting angle of the bulk utilizing numerical tests with the natural resting angle in the PFC^2D^ linear contact model and concluded that the natural resting angle tends to increase first and then stabilize as the friction and rotation coefficients increase. Zhou et al. [16] carried out many planar biaxial compression tests on cohesive soil-like material samples with the aid of the particle discrete element analysis software PFC^2D^. The shear strength parameters (internal friction angle, cohesion) of the numerical specimens were calibrated by recording the peak axial stresses of the samples under different confining pressures and according to the Mohr–Coulomb strength criterion. It is noted that the particle bond (normal and tangential) strength is linearly related to the cohesion of the material. The particle friction coefficient is approximately logarithmically related to the internal friction angle of the material. The particle stiffness ratio also has a weak effect on the variation of the material shear strength parameters. In addition, the *K* value (ratio of tangential bond strength to normal bond strength) is an important factor affecting the shear damage pattern of the material. Wu et al. [17] applied a new brittle cluster parallel-bond model to consider the strong occlusion of irregular mineral grains in brittle granites to compensate for the problem of too small tensile to the compressive ratio in the parallel-bond model. This new method allows the simulation of the high strength ratio (ratio of uniaxial compressive strength to tensile strength) and the brittle fracture characteristics of granite. Potyondy et al. [18] found in the macroscopic study that simulating rocks with PBM would result in too low a compression-tension ratio, and even if the cohesive ratio of the particles was changed, the compression-tension ratio of the actual rock could not be met, and the simulated internal friction angle of the rock was small. Based on the shortcomings of this model, a flat nodal model is proposed. In the flat nodal model setting, the original circular particle structure is assumed to be a polygonal particle structure, and under this contact model with an assumed polygonal particle structure, an internal locking effect can be generated between the particles, thus inhibiting the rotation of the bonded particles after disruption, and thus improving the tensile pressure ratio. The PFC^2D^ numerical simulations of uniaxial compression and Brazilian splitting of rocks were investigated by Liu et al. [19] using the flat- joint model (FJM). Chen et al. [20] have used the PFC^2D^ flat-nodal model (FJM) to investigate and calibrate micro and macro mechanical parameters. Su et al. [21] used the flat-joint model (FJM) to investigate the macroscopic strength effects of microscopic strength coefficients of variation affecting rocks and the ease of crack generation and obtained macroscopic equations between the microscopic coefficients and macroscopic mechanical parameters (elastic parameters, Poisson’s ratio, uniaxial compression strength, and crack gap stress to uniaxial compression strength ratio). Chen et al. [22] investigated the calibration of the fine-scale parameters of the uniaxial compression PFC^2D^ model for rocks. Bahaaddini et al. [23] used the flat-joint model (FJM) to investigate the effect of the microscopic parameters flat-joint adhesion ratio on the macroscopic parameters (elastic modulus, Poisson’s ratio and compression-tension ratio) of rocks. Li et al. [24] investigated the effect of microscopic coefficients on macroscopic coefficients of the parallel bond model based on PFC^3D^ and established quantitative equations for macroscopic and microscopic parameters. Tan et al. [25] proposed a new method for calibrating PFC^3D^ fine-scale parameters considering fracture toughness. Feng et al. [26] used PFC^2D^ to calibrate the macroscopic parameters (uniaxial compressive strength, uniaxial tensile strength, Poisson’s ratio and modulus of elasticity) by the trial-and-error method, and the damage modes of the test blocks were also calibrated. Deng et al. [27] calibrated the fine-scale parameters in the numerical simulation study of hard rock masses and concluded that the uniaxial compressive strength and Brazilian splitting strength of rocks were mainly influenced by the tangential and normal strength of the adhesion, and the greater the adhesion strength, the greater the macroscopic tensile strength of rocks; the elastic modulus of rocks was mainly influenced by the fine-scale elastic modulus and stiffness ratio, where the fine-scale elastic modulus has a positive influence on the macroscopic elastic modulus. The stiffness ratio mainly affects the Poisson’s ratio of the rock, and the larger the stiffness ratio of the rock, the larger the Poisson’s ratio of the rock. Zhang et al. [28] investigated the correlation between the eight microscopic parameters of the flat-joint model and the six macroscopic parameters of the rock using orthogonal numerical tests by PFC^2D^, determined the fitting relationships between each macroscopic parameter and the main microscopic parameters, and analysed the trend relationships between the macroscopic parameters. Hao et al. [29] carried out uniaxial compression tests, uniaxial tensile tests, and triaxial compression tests by PFC^2D^, and used orthogonal tests to derive the relationships between the macroscopic parameters (rock modulus of elasticity, Poisson’s ratio, uniaxial compressive strength, initiating crack strength, tensile to compressive strength ratio and friction angle) and the microscopic parameters.

The process of determining the mechanical properties of rocks in indoor tests often uses oil pressure loading, whereas the above authors have used rigid wall boundary loading to analyze the PFC^3D^ macroscopic relationships. However, in the process of numerical simulation, the loading of rigid and flexible boundaries makes the crack development of the rock and the results of the damage pattern of the rock mass different, so different boundary loading methods will have some influence on the results of our study of microscopic damage patterns. Secondly, the contact strength distribution in the studies of some authors mentioned above using a uniform distribution does not reflect the discrete nature of their materials, and the contact strength distribution of the rock is also related to the cracking strength of the rock. Therefore, considering the influence of the above factors, in the study of this paper, orthogonal experiments are used to simulate the flexible boundary by a series of boundary particles constituted by the membrane particle boundary, and the contact intensity distribution is adopted as Gaussian distribution to explore the relationship of PFC^3D^ macrofine view parameters, aiming to provide a reference basis for the parameter calibration under this condition. Finally, this paper obtains the correlation of macroscopic variables by regression statistical analysis of the relationship of macroscopic variables, and obtains the macroscopic equation based on the correlation fitting to give a parameter calibration process based on Gaussian intensity distribution under the flexible boundary.

## 2. The Basic Principle of PFC3D

### 2.1. Flat-Joint Model

The flat-joints model (FJM) is made up of grains and intergranular endowed with flat joints contact (FJC), where the grains consist of disc particles and notional surfaces, as shown in Figure 1 below. In the flat-nodal model setup, the original circular particle configuration is assumed to be polygonal particles. Under this contact model, with assumed polygonal particles, an internal locking effect can be generated between the particles, thus inhibiting the rotation of the bonded particles after destruction and thus improving the tensile pressure ratio. There are two states of adhesion and unadhesion on the contact selection of the flat-nodal model. For the unadhesive part, (1) when $\hat{\sigma }<0$, $\tau_{b}=-\mu_{b}\hat{\sigma }$; (2) when $\hat{\sigma }\geq 0$, $\tau_{b}=$ 0. If $\left|{\hat{\tau }}^{\prime }\right|\leq \tau_{c}$, then, the shear strength of the contact particles is ${\hat{\tau }}^{\prime }$; otherwise, the particles will slide against each other, and the shear strength of the contact particles is ${\hat{\tau }}^{\prime }\left(\frac{\tau_{c}}{{\hat{\tau }}^{´}}\right)$. For the adhesive part, $\tau_{b}=c_{b}-\hat{\sigma }\tan\varphi_{b}$. If $\left|{\hat{\tau }}^{´}\right|\leq \tau_{c}$, the shear strength of the contact particles is ${\hat{\tau }}^{´}$; otherwise, the particles will undergo shear failure, resulting in shear cracks. If $\hat{\sigma }>\sigma_{b}$, tensile damage occurs in the bond, resulting in tensile cracks; where *σ*_b_ is the tensile strength. In this paper, the flat-nodal model contact in the cohesive state is chosen, and a different cohesive radius is given to consider the denseness of its particles.

### 2.2. Selection of Rock Macro and Micro Parameters

Most scholars [19,20,21,22] have chosen the modulus of elasticity *E*, Poisson’s ratio *ν*, and the uniaxial compressive strength *UCS* to calibrate the microscopic parameters of the model and then carry out numerical analysis. Chen et al. [20] pointed out that the model obtained by using only these parameters as calibration indicators cannot be used for the properties of rocks under multi-directional stress state, so this paper considers these indicators based on the addition of rock tensile strength and the strength indicators *c* and *φ* under peritectic pressure. The strength indexes *c* and *φ* under compression can be expressed by *UCS* and *φ*; see Equation (1). Therefore, in this paper, the microscopic parameters are selected as *E*_ƒ_, *k*_ƒ_, *μ*_ƒ_, *σ*_ƒ_, *C*_ƒ_, *φ_f_*, *R*_sd_, *θ*_b_. A summary of the macroscopic parameter selections is given in Table 1.
(1)$$C=\frac{UCS(1-\sin\phi )}{2\cos\phi }$$

### 2.3. Establishment of Numerical Rock Simulation Experiments

Su et al. [21] showed that when the ratio of model height to mean particle radius *L*/*R* ≥ 125, the particle size does not affect the macroscopic parameters. Zhou et al. [11] showed that when (*L*/*R*_min_) [1/(1 + *R*_max_/*R*_min_)] ≥ 10, the size and number of particles have less influence on the macroscopic mechanical parameters of the model, where *L* is the minimum scale of the model and *R*_min_ and *R*_max_ are the minimum and maximum diameters, respectively. Potyondy et al. [18] suggested that *R*_max_/*R*_min_ = 1.66 without considering the gradation so that the generated rock is more consistent with the physical properties of the rock. Therefore, in this paper, the particle size in the numerical simulation specimen is chosen as *R*_max_/*R*_min_ = 1.66 (*R*_min_ = 0.8 mm) and the particle density is 2600 kg/m^3^, which meets the above requirements. Numerical simulations of rocks were used to compare the results of indoor experiments for the calibration of macroscopic parameters. The tensile strength of rocks is generally measured indirectly by Brazilian splitting. The uniaxial tensile strengths measured by direct tensile numerical simulations differed from those measured by indoor Brazilian splitting experiments, so the Brazilian splitting numerical simulations were chosen to calibrate the uniaxial tensile strengths measured by macroscopic Brazilian splitting indoor experiments. A cylindrical specimen of the same size with a diameter of 50 mm and a height of 25 mm was established for the numerical simulation according to the size of the specimen for the indoor experiments, as shown in Figure 2. The right side of the figure represents the actual loading direction of the Brazilian splitting.

The compressive strength of the rock was calibrated utilizing uniaxial compression simulations established by numerical simulation experiments. The dimensions of the numerically simulated specimens are the same as those of the uniaxial compression specimens in the chamber, both being cylindrical specimens with a diameter of 50 mm and a height of 100 mm, as shown in Figure 3. The internal friction angle and cohesion of the rock were measured from the indoor triaxial experiments, and the numerical simulations also created cylindrical specimens of the same size as the indoor triaxial specimens, with a diameter of 50 mm and a height of 100 mm. As the triaxial indoor specimens were loaded by confining pressure, the rigid walls that applied the confining pressure were replaced by flexible membrane particles in the numerical simulations, as shown in Figure 4. A linear contact model was used to contact the membrane particles, which better characterises the flexible membrane. The red particles in the middle represent membrane particles, and the blue and green particles above and below represent boundary particles. The right side of Figure 3 and Figure 4 respectively represents the actual loading direction.

## 3. Orthogonal Experimental Design and Analysis of Results

### 3.1. Selection of Factor Levels for Orthogonal Experiments

Because of the large number of micro variables selected for study in this paper, the control variables method is more troublesome to study the effect of multiple factors. The orthogonal experimental design is another method to learn multi-factor and multi-level, and based on orthogonality, selects some representative points from the comprehensive test. These representative points have the characteristics of being ‘uniformly dispersed, flush, and comparable’. The orthogonal experimental design is the primary method to analyze the factorial design. In this paper, the selection of *E*_ƒ_, *k*_ƒ_, *μ*_ƒ_, *σ*_ƒ_, *C*_ƒ_, *φ_f_*, *R*_sd_, *θ*_b_ and other microscopic parameters as factor levels for the orthogonal experiments are detailed in Table 2 below. At the same time, a reasonable range is selected based on the previous research results, and it is verified that within this range, elastic parameters, Poisson’s ratio, uniaxial compressive strength, crack initiation strength, uniaxial tensile strength, and internal friction angle all cover the value range of soft and hard rock, which is more reasonable for rock calibration.

### 3.2. Establishment of Orthogonal Experimental Table and Analysis of Results

An orthogonal numerical simulation table was established according to the factor levels obtained above. The macroscopic parameters (elastic modulus, Poisson’s ratio, compressive strength, cracking strength, tensile strength, and internal friction angle) were determined by Brazilian splitting, uniaxial compression, and flexible triaxial numerical simulation. The results of the orthogonal experiments are shown in Table 3 below.

#### 3.2.1. Multi-Factor Analysis of Variance

Because each micro parameter in the PFC has a different degree of influence on each macro parameter, in order to analyze the magnitude of the effect of each micro parameter on the macro parameter, an analysis of variance (ANOVA) was performed on the micro variables by SPSS software. The *F*-value (Equality of Variances) of each micro parameter was calculated as follows in Figure 5.

In the ANOVA, the micro variables of the multifactorial PFC were analyzed for significance using the *F*-test (joint hypotheses test). From the table, *F* = 4.35 when the significance level *a* = 0.05 and *F* = 8.45 when the significance level *a* = 0.01. The *F* values of each part of micro parameters were calculated according to the orthogonal experiment table, and it was concluded that when 4.35 < *F* < 8.45, micro factors significantly affect macro parameters. When *F* > 8.45, the micro factors significantly affect the macro parameters, and the larger the *F* value, the greater the significance effect.

Among the macroscopic parameters of elastic modulus, the *F*-value of the flat-joint elastic modulus, flat-joint stiffness ratio, and the flat-joint tensile strength *σ_ƒ_* are 442.18, 179.53, and 32.6, respectively, indicating that these microscopic parameters have a remarkable effect on the macroscopic parameter elastic modulus and the magnitude of the value represents the difference in the degree of influence accounted for. The flat-joint bond strength *C*_ƒ_, flat-joint adhesion ratio *θ*_b_, flat-joint adhesion strength coefficient of variation *R*_sd_, flat-joint friction coefficient *μ*_ƒ_, and flat -oint friction angle *φ_f_*, which have F-values less than 4.35, are insignificant. The meanings of the microscopic parameters with considerable significance are close to those of the macroscopic parameters, which are informative.

Among the macroscopic parameters of Poisson’s ratio, the *F*-value of the flat- joint stiffness ratio *k*_ƒ_, *F*-value of the flat-joint modulus of elasticity *E*_ƒ_, and flat-nodal bond strength *C*_ƒ_ are 116.01, 8.54, and 23.86, respectively, indicating that these microscopic parameters have a significant effect on the macroscopic parameter of Poisson’s ratio. The flat-joint friction coefficient *μ*_ƒ_, flat -oint adhesion ratio *θ*_b_, flat-joint adhesion strength coefficient of variation *R*_sd_, flat-joint tensile strength *σ*_ƒ_, and flat-joint friction angle *φ_f_* all have F-values less than 4.35, which are not significant. The meanings of the microscopic parameters with immense significance are close to those of the macroscopic parameters, which are informative.

Among the macroscopic parameters of compressive strength, the *F*-value of the flat-joint bond strength *C*_ƒ_, flat-joint tensile strength *σ*_ƒ_, and flat-joint stiffness ratio *k*_ƒ_ are 42.25, 8.63, and 23.27, respectively, indicating that these microscopic parameters have a significant effect on the macroscopic parameter compressive strength. The *F* value of 5.23 for the flat-joint modulus of elasticity *E*_ƒ_ indicates a significant effect of this microscopic parameter, but the significance is not great when comparing the first three microscopic factors. The remaining F-values of flat-joint adhesion ratio *θ*_b_, flat-joint adhesion strength coefficient of variation *R*_sd_, flat-joint friction coefficient *μ*_ƒ_, and flat-joint friction angle *φ_f_* were less than 4.35, which were not significant.

Among the macroscopic parameters of tensile strength, the *F*-value of 120.71 for the flat-joint tensile strength *σ_ƒ_* and 43.64 for the flat-joint stiffness ratio *k_ƒ_* indicate that these microscopic parameters have a significant effect on the macroscopic parameters of tensile strength taking values, whereas the *F* value of 8.6 for the flat-joint modulus of elasticity *E*_ƒ_ indicates that these microscopic parameters have a significant effect on the macroscopic parameters’ tensile strength. The *F* value of 5.5 for the flat-joint bond strength *C*_ƒ_ indicates that this microscopic parameter has a significant effect, but the significance is not great when comparing the first three microscopic factors. The remaining *F*-values of flat-joint friction coefficient *μ*_ƒ_, flat-joint adhesion ratio *θ*_b_, flat-joint adhesion strength coefficient of variation *R*_sd_, and flat-joint friction angle *φ_f_* were less than 4.35, which were not significant.

In the macroscopic parameter of internal friction angle, the *F*-value of flat-joint modulus of elasticity *E*_ƒ_, the flat-joint stiffness ratio *k_ƒ_* of 14.86, the flat-joint friction coefficient *μ_ƒ_*, and the flat-joint friction angle *φ_f_* are 9.98, 14.86, 13.09, and 8.9, respectively, indicating that these microscopic parameters have a significant effect on the value of the macroscopic parameter of friction angle. The remaining F-values of flat-joint adhesion ratio *θ*_b_, flat-joint bond strength *C_ƒ_*, and flat-joint adhesion strength coefficient of variation *R*_sd_ are less than 4.35, which are not significant.

Among the macroscopic parameters of crack initiation strength, the *F*-value of the coefficient of variation of flat-joint adhesion strength *R*_sd_ is 26.98, which indicates that this microscopic parameter has a significant effect on the value of macroscopic parameters of crack initiation strength. The *F*-values of 7.17 for the flat-joint tensile strength *σ*_ƒ_ and 6.03 for the flat-joint adhesion ratio *θ*_b_ indicate a significant effect of these microscopic parameters, but the significance is not significant compared to the coefficient of variation of the flat-joint adhesion strength *R*_sd_.

#### 3.2.2. Multi-Factor Regression Analysis

The magnitude of the influence of each microscopic parameter in each macroscopic mechanical parameter was obtained by the above multi-factor ANOVA, and the data from the orthogonal experimental results were averaged as in Figure 6. The macroscopic equations were fitted by the study of the positive and negative correlations of each microscopic parameter with the macroscopic parameters.

The means of the analysis of the variance of elastic modulus reveals that the flat-joint elastic modulus *E*_ƒ_, flat-joint stiffness ratio *k*_ƒ_, and flat-joint tensile strength *σ*_ƒ_ have a significant effect on the macroscopic parameter elastic modulus. In the correlation analysis, where the flat-joint modulus of elasticity *E*_ƒ_ is positively correlated with the macroscopic parameter modulus of elasticity, the flat-joint stiffness ratio *k*_ƒ_ is negatively correlated with the macroscopic parameter modulus of elasticity, and the flat-joint tensile strength *σ*_ƒ_ is positively correlated with the macroscopic parameter modulus of elasticity, which is negatively correlated.

The analysis of Poisson’s ratio variance is known in which the flat-joint stiffness ratio *k*_ƒ_, flat-joint modulus of elasticity *E*_ƒ_, and flat-joint bond strength *C*_ƒ_ have a significant effect on the macroscopic parameter of Poisson’s ratio. In the correlation analysis, where the flat-joint stiffness ratio *k*_ƒ_ is positively correlated with the macroscopic parameter of Poisson’s ratio, the flat-joint modulus of elasticity *E*_ƒ_ is negatively correlated with the macroscopic parameter of Poisson’s ratio, and the flat-joint bond strength *C*_ƒ_ is positively correlated with the macroscopic parameter of Poisson’s ratio.

In the analysis of variance of compressive strength, it can be concluded that among them, the flat-joint bond strength *C*_ƒ_, the flat-joint tensile strength *σ*_ƒ_, and the flat-joint stiffness ratio *k*_ƒ_ have a significant effect on the macroscopic parameter of compressive strength. In the correlation analysis, the flat-joint bond strength *C*_ƒ_ positively correlates with the macroscopic compressive strength parameter. The flat-joint tensile strength *σ*_ƒ_ is positively correlated with the macroscopic parameter compressive strength, and the flat-joint stiffness ratio *k*_ƒ_ is negatively correlated with the macroscopic parameter of compressive strength.

The analysis of variance of the modulus of elasticity is known for the flat-joint tensile strength *σ*_ƒ_ and flat-joint stiffness ratio *k*_ƒ_, indicating that these microscopic parameters significantly affect the values taken for the macroscopic parameter of tensile strength. In the correlation analysis, the flat-joint tensile strength *σ*_ƒ_ is positively correlated with the macroscopic parameter of tensile strength. The flat-joint stiffness ratio *k*_ƒ_ is negatively correlated with the macroscopic parameter of tensile strength.

The analysis of variance of the friction angle shows that the flat-joint modulus of elasticity *E*_ƒ_, flat-joint stiffness ratio *k_ƒ_*, flat-joint friction coefficient *μ*_ƒ_, and flat-joint friction angle *φ_f_* have significant effects on the values taken for the macroscopic parameter of friction angle. In the correlation analysis, the flat-joint modulus of elasticity *E*_ƒ_ is positively correlated with the macroscopic parameter of friction angle. The flat-joint stiffness ratio *k*_ƒ_ is positively correlated with the macroscopic parameter of friction angle. The flat-joint friction coefficient *μ*_ƒ_ is positively correlated with the macroscopic parameter of friction angle. The flat-joint friction angle *φ_f_* is positively correlated with the macroscopic parameter of friction angle.

The coefficient of variation of the flat-joint bond strength, *R*_sd_, the flat-joint tensile strength *σ_ƒ_*, and the flat-joint bond ratio *θ*_b_, is found to have significant effects on the values of the macroscopic parameter of crack initiation strength in the ANOVA of crack initiation strength. In the correlation analysis, the coefficient of variation of flat-joint bond strength *R*_sd_ is negatively correlated with the macroscopic parameter of cracking strength; the flat-joint tensile strength *σ*_ƒ_ is positively correlated with the macroscopic parameter of cracking strength; and the flat joint bonding ratio *θ*_b_ is negatively correlated with the macroscopic parameter of cracking strength.

### 3.3. Fitting of Macro and Micro Equations

After the above ANOVA and correlation analysis, the micro parameters corresponding to their maximum influence amount were obtained for each macro parameter. A linear regression statistical analysis was performed using SPSS software to perform a multi-factor linear fit to the data in the orthogonal experiment, and the results of the linear fit are detailed in Table 4 below. The preliminary solutions obtained by linearly fitting the equations will indeed have different combinations of microscopic parameters, and we need to carry out the microscopic parameter solutions for selecting the same damage mode as the indoor experiments, so some values will be fixed during the calibration process to achieve a damage mode that matches the actual one, such as the flat nodal strength ratio. As can be seen from the table below, only the elastic parameters, Poisson’s ratio, uniaxial compressive strength, and uniaxial tensile strength have good fits. In contrast, the corresponding fits for the internal friction angle and cracking strength are low, and the number of microscopic coefficients is greater than the number of macroscopic solutions.

## 4. Example of Rock Calibration Procedure Verification

Rock calibration examples were chosen to obtain macro-mechanical parameters from physical experiments on Transjuane Sandstone [30,31]. The experimental averages of macroscopic mechanical parameters for soft rocks are shown in Table 5.

Simple linear fits are better for modulus of elasticity, Poisson’s ratio, compressive strength and tensile strength, and the corresponding values were calculated with a small error rate. In comparison, linear fits for cracking stress and internal friction angle are lower and have too many parameters, so the flat joint adhesion ratio *θ*_b_ is determined based on previous experience and the denseness of the rock. The microscopic coefficient flat-nodal cohesion ratio *θ*_b_ is the cohesion distance between the centres of the microscopic particles, according to the soft rock’s denseness, so the cohesion radius value is taken as 0.6. The above fitting formula calculates the remaining parameters, and the modulus of elasticity *E*_ƒ_ initially calculates the effective modulus of the microscopic parameter of flat joint. The Poisson’s ratio *k*_ƒ_ initially calculates the stiffness ratio of the microscopic parameter of flat joint. The cohesion of the flat joint is initially calculated by the compressive strength *C*_ƒ_. The tensile strength of flat joints is calculated initially from the tensile strength *σ*_ƒ_. The coefficient of friction of flat joints is calculated from the angle of internal friction *μ*_ƒ_ and the angle of friction of flat joints *φ_f_*. The values of the microscopic parameters taken above are brought into uniaxial numerical simulation experiments, flexible triaxial simulation experiments, and Brazilian splitting simulation experiments to obtain the preliminary calibrated macroscopic parameters.

The damage mode determined by the relative ratio of the calculated cohesive force *C*_ƒ_ of the flat joints to the tensile strength *σ*_ƒ_ of the flat joints can be seen from previous studies. As shown in Figure 7 below, the penetration cracks in the indoor Brazilian splitting experiments are in good agreement with those in the numerical simulations. The tension cracks in the numerical simulation are also similar to the splitting tension damage in the indoor experiments. Where different colored particles represent broken particles. As can be seen in Figure 8 below, the penetration cracks in the numerical simulation of single-period compression are more compounded with the penetration cracks in the actual indoor tests; therefore, the relative ratio of the microscopic parameter flat joint cohesion *C*_ƒ_ to the flat joint tensile strength *σ*_ƒ_ is reasonable based on the values chosen on the damage model.Initial determination of microscopic parameters are shown in Table 6. According to Figure 9, Figure 10 and Figure 11, it can be seen that the uniaxial compression curves are similar to the actual indoor test curves, and the curves of the flexible boundary triaxial tests also meet the curve requirements of the actual indoor tests. The rock sample of the indoor experiment has original cracks and pores, and with the loading of force, there will be a compression-density process, whereas the pre-pressure process of the specimen is already available in the early stage of the numerical simulation process. After removing the compression-density stage of the indoor experiment, the curve of the indoor experiment and the simulation curve can be better fitted, i.e., the indoor loading curve is modified to start loading from the linear elastic stage. The curves of the Brazilian splitting experimental results also reflect the apparent splitting characteristics, so the microscopic parameters selected according to the calibration formula are reasonable based on the macroscopic mechanical properties.

Table 7 shows that, except for Poisson’s ratio and internal friction angle, which have larger error rates, the rest of the calibrated macro parameters are within a reasonable error of 5 per cent, thus proving that the fitting formula is more reasonable in the calibration process. Then, the elastic modulus needs to be corrected by fine-tuning of the flat-joint effective modulus *E*_ƒ_. The Poisson’s ratio is corrected by fine-tuning the flat-nodular stiffness ratio *k*_ƒ_. The tensile strength can be obtained by fine-tuning the flat-joint tensile strength *σ*_ƒ_. Uniaxial compressive strength is obtained by fine-tuning the flat-joint cohesion force *C*_ƒ_. The internal friction angle is fine-tuned by fine-tuning the flat-nodal friction coefficient *μ_ƒ_* and the flat-nodal friction angle *φ_f_*. The final values obtained are shown in the following table (Table 8 and Table 9). In this paper, a general calibration process is summarized by the above macroscopic parameter study, (1) selecting the appropriate flat-joint adhesion radius according to the density; (2) achieving the damage phenomenon consistent with the indoor experiment by selecting the flat-joint strength tensile compression ratio; (3) calculating the remaining microscopic parameters needed, according to the equation; and (4) fine-tuning according to the obtained values to finally meet the error range. The flow chart is detailed in Figure 12.

## 5. Conclusions

(1)The study of PFC^3D^ macro and micro parameters based on the flexible boundary, from multi-factor ANOVA and orthogonal experimental averages, obtained each microscopic parameter (flat-joint effective modulus *E*_ƒ_, flat-joint stiffness ratio *k*_ƒ_, flat- joint friction coefficient *μ*_ƒ_, flat-joint tensile strength *σ*_ƒ_, flat-joint bond strength *C*_ƒ_, flat-joint friction angle *φ_f_*, flat-joint bond ratio *θ*_b_, and coefficient of variation of flat- nodal bond strength *R*_sd_) and macroscopic parameter (modulus of elasticity *E*, Poisson’s ratio *v*, uniaxial compressive strength *UCS*, crack initiation strength *σ*_ci_, internal friction angle *φ*, and uniaxial tensile strength *TS*). Then, each microscopic parameter is obtained as a function of the macroscopic mechanical parameters, and the macroscopic equations are obtained by multifactor linear fitting. It is shown that the fitting equation is reasonable, and it is more efficient than the trial-and-error method when using the fitting equation for the initial selection and then fine-tuning according to the significance analysis and analysis of variance when considering the calibration of multiple macro-mechanical parameters.(2)The improvement of the rigid boundary makes the numerical simulation and the indoor test more realistic. Considering the confinement of the rock in the triaxial test and from the results of the triaxial numerical simulation, the triaxial stress values calibrated by the numerical simulation of the rigid wall are all larger than those of the flexible boundary because the lateral constraint of the rigid wall is stronger than that of the flexible film particles.(3)In the calibration process, the macroscopic properties of the rocks are considered, but the damage mode is also calibrated by a reasonable value of the relative ratio of the cohesive force *C*_ƒ_ to the tensile strength *σ*_ƒ_ of the flat joints. Then, the rock is calibrated for microscopic crack appearance and stable to unstable crack development by taking reasonable values for crack initiation stress, which is helpful for the subsequent study of the rock in the damage model and the crack development mechanism.(4)In the calibration process, the composition of different rocks is considered by the ratio of flat-joint adhesion *θ*_b_, and the appropriate ratio of flat-joint adhesion *θ*_b_ is selected for different rocks with different densities. From the significance analysis and ANOVA, it is clear that this parameter has a negligible effect on the macroscopic mechanical properties so that it can be determined according to the actual rock composition properties during the calibration process.

## Figures and Tables

**Figure 1 materials-15-06790-f001:**
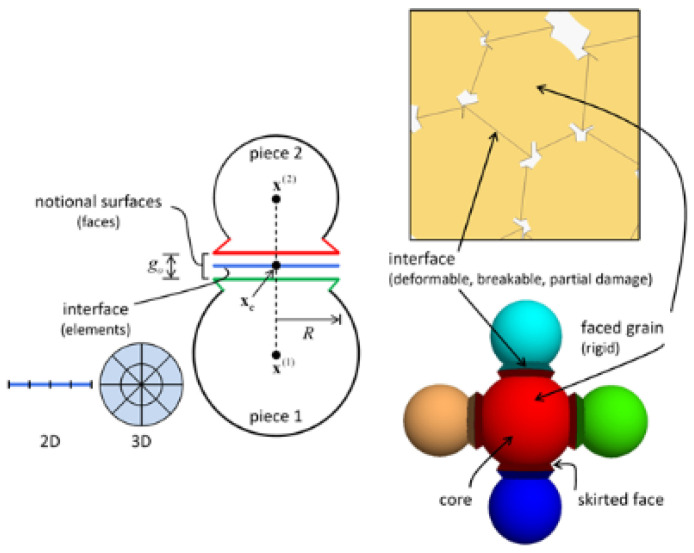
Flat-joint contact mode.

**Figure 2 materials-15-06790-f002:**
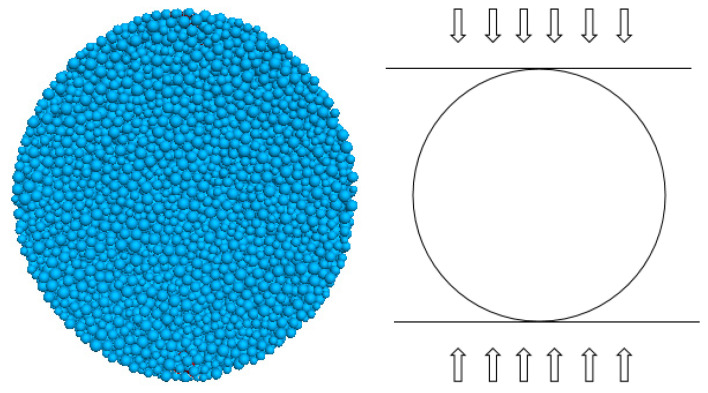
Numerical simulation of Brazilian splitting.

**Figure 3 materials-15-06790-f003:**
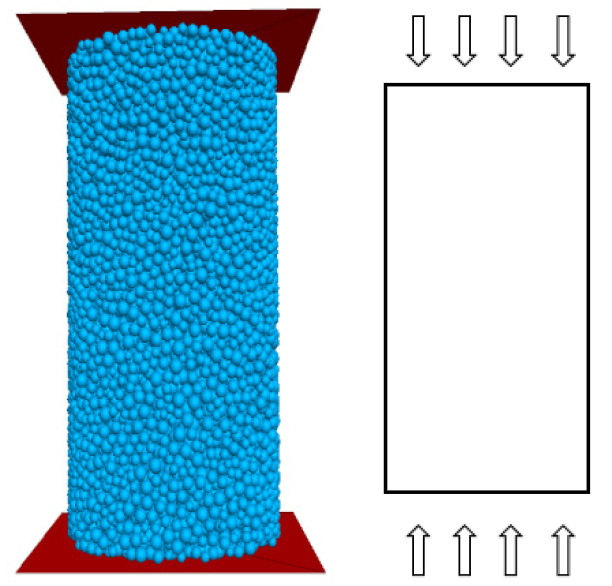
Numerical simulation of uniaxial compression.

**Figure 4 materials-15-06790-f004:**
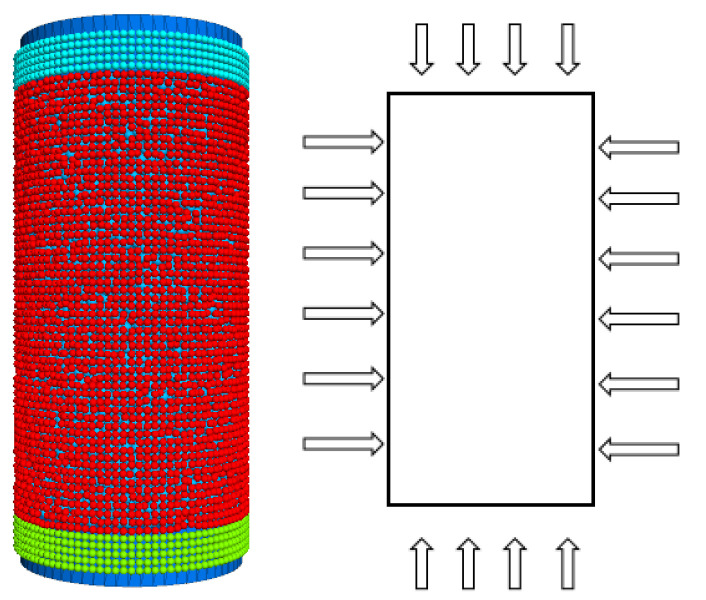
Triaxial numerical simulation experiment.

**Figure 5 materials-15-06790-f005:**
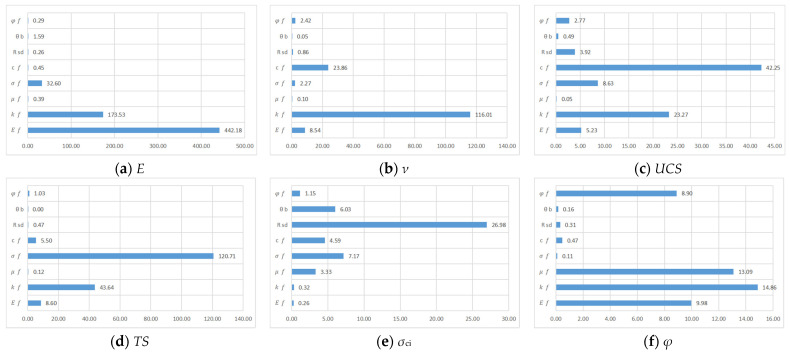
Analysis of variance F value of each microscopic parameter.

**Figure 6 materials-15-06790-f006:**
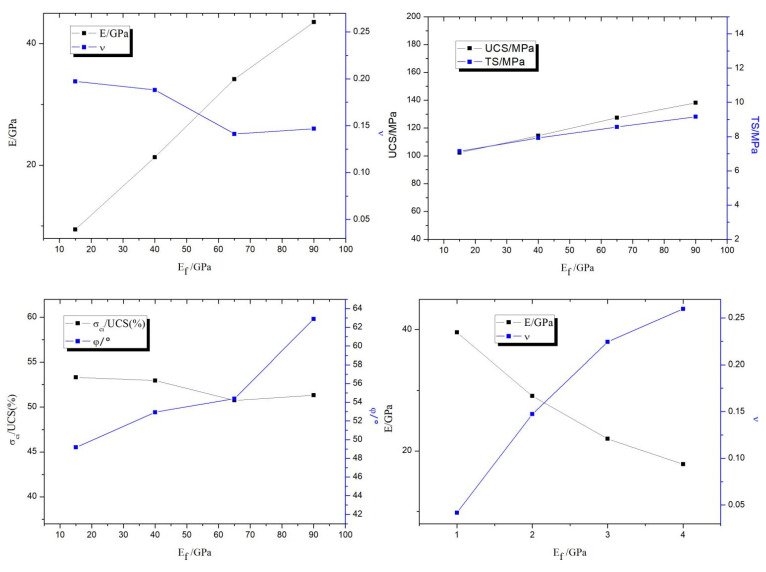
Average value of microscopic parameter results.

**Figure 7 materials-15-06790-f007:**
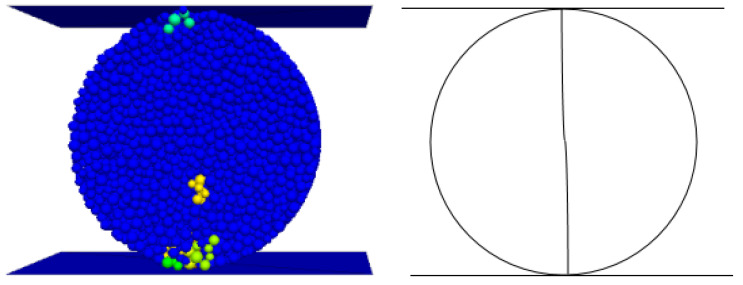
Transjuane Sandstone specimen after splitting failure in Brazil.

**Figure 8 materials-15-06790-f008:**
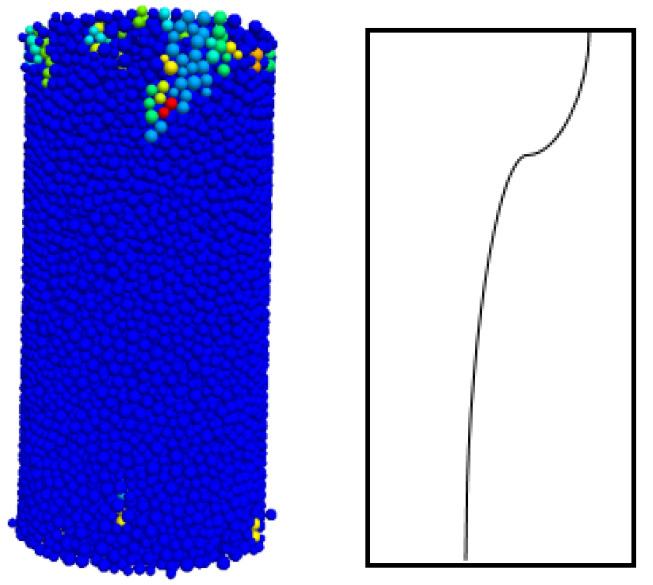
Transjuane Sandstone specimen after uniaxial failure.

**Figure 9 materials-15-06790-f009:**
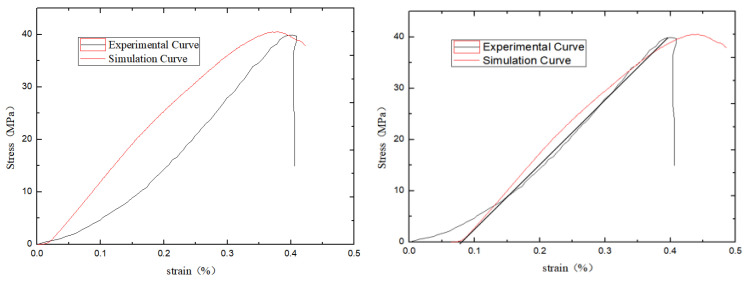
Comparison of experimental data and numerical simulation under uniaxial compression.

**Figure 10 materials-15-06790-f010:**
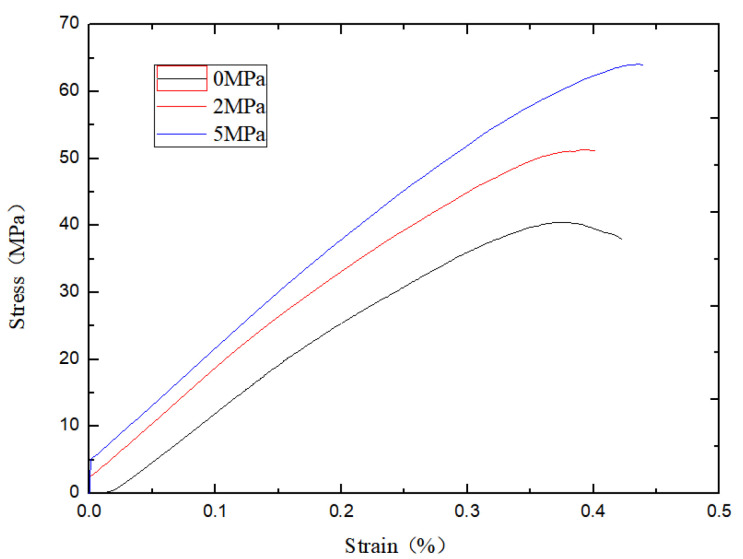
Numerical simulation triaxial experiment.

**Figure 11 materials-15-06790-f011:**
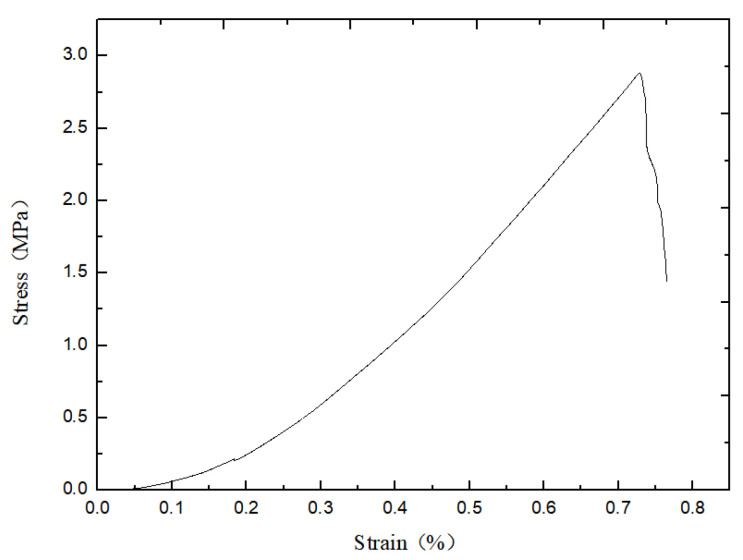
Numerical simulation of Brazilian splitting experiment.

**Figure 12 materials-15-06790-f012:**
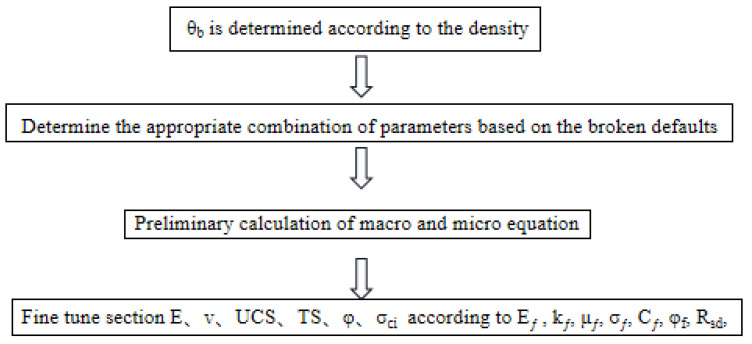
Calibration flow chart.

**Table 1 materials-15-06790-t001:** The selected micro and macro parameters.

Micro-Parameter	Macro-Parameter
*E* _ƒ_	*E*
*k* _ƒ_	*ν*
*μ* _ƒ_	*UCS*
*σ* _ƒ_	*TS*
*C* _ƒ_	*φ*
*φ_f_*	*σ* _ci_
*R* _sd_	
*θ* _b_	

**Table 2 materials-15-06790-t002:** Factor levels.

Factor Levels	*E*_ƒ_/GPa	*k* _ƒ_	*μ* _ƒ_	*σ*_ƒ_/MPa	*C*_ƒ_/*σ*_ƒ_	*φ*/°	*R* _sd_	*θ* _b_
1	15	1	0.1	6	2	0	0.1	0.7
2	40	2	0.4	14	5	20	0.25	0.8
3	65	3	0.7	22	8	40	0.4	0.9
4	90	4	1	30	11	60	0.5	1

**Table 3 materials-15-06790-t003:** Orthogonal numerical test scheme and results.

Number	Micro-Parameter								Macro-Parameter					
	*E*_ƒ_/GPa	*k* _ƒ_	*μ* _ƒ_	*σ*_ƒ_/MPa	*C_ƒ_*/*σ_ƒ_*	*φ*_ƒ_/°	*R* _sd_	*θ* _b_	*E*/GPa	*ν*	*UCS*/ MPa	*TS*/MPa	*σ*_ci_/UCS(%)	*φ*/°
1	15.00	1.00	0.10	6.00	2.00	0.00	0.1	0.7	12.32	0.028	14.78	1.22	77.60	13.19
2	15.00	2.00	0.40	14.00	5.00	20.00	0.25	0.8	10.43	0.155	88.12	6.91	67.32	51.89
3	15.00	3.00	0.70	22.00	8.00	40.00	0.4	0.9	7.22	0.288	136.74	8.81	45.12	64.76
4	15.00	4.00	1.00	30.00	11.00	60.00	0.55	1.0	5.58	0.375	158.40	10.33	35.69	67.10
5	40.00	1.00	0.10	14.00	5.00	40.00	0.4	1.0	32.82	0.041	94.13	8.94	50.39	34.77
6	40.00	2.00	0.40	6.00	2.00	60.00	0.55	0.9	20.08	0.108	24.98	1.74	30.90	63.79
7	40.00	3.00	0.70	30.00	11.00	0.00	0.1	0.8	18.55	0.348	247.83	12.98	56.18	60.91
8	40.00	4.00	1.00	22.00	8.00	20.00	0.25	0.7	10.13	0.337	135.43	7.93	53.83	61.58
9	65.00	1.00	0.40	22.00	11.00	20.00	0.1	0.9	53.39	0.041	292.68	13.54	46.75	46.94
10	65.00	2.00	0.10	30.00	8.00	0.00	0.25	1.0	40.74	0.135	201.25	14.73	54.45	39.00
11	65.00	3.00	1.00	6.00	5.00	60.00	0.4	0.7	21.61	0.180	43.55	2.55	42.00	70.38
12	65.00	4.00	0.70	14.00	2.00	40.00	0.55	0.8	24.99	0.176	42.40	3.70	39.78	62.18
13	90.00	1.00	0.40	30.00	8.00	60.00	0.4	0.8	67.44	0.042	378.92	21.40	55.69	59.24
14	90.00	2.00	0.10	22.00	11.00	40.00	0.55	0.7	45.46	0.153	174.87	10.22	43.03	61.84
15	90.00	3.00	1.00	14.00	2.00	20.00	0.1	1.0	40.98	0.114	53.31	5.36	65.82	69.07
16	90.00	4.00	0.70	6.00	5.00	0.00	0.25	0.9	29.31	0.132	31.28	2.03	54.34	66.66
17	15.00	1.00	1.00	6.00	11.00	40.00	0.25	0.8	13.99	0.060	105.23	4.62	42.71	47.04
18	15.00	2.00	0.70	14.00	8.00	60.00	0.1	0.7	9.89	0.220	149.23	7.65	74.36	56.80
19	15.00	3.00	0.40	22.00	5.00	0.00	0.55	1.0	7.87	0.226	86.86	8.17	35.23	51.14
20	15.00	4.00	0.10	30.00	2.00	20.00	0.4	0.9	8.06	0.226	78.61	9.43	48.47	41.42
21	40.00	1.00	1.00	14.00	8.00	0.00	0.55	0.9	29.62	0.055	135.22	9.27	32.12	27.51
22	40.00	2.00	0.70	6.00	11.00	20.00	0.4	1.0	21.06	0.168	62.24	3.05	38.47	59.87
23	40.00	3.00	0.40	30.00	2.00	40.00	0.25	0.7	23.19	0.164	101.46	11.36	74.95	56.14
24	40.00	4.00	0.10	22.00	5.00	60.00	0.1	0.8	15.21	0.284	114.69	8.08	86.92	58.90
25	65.00	1.00	0.70	22.00	2.00	60.00	0.25	1.0	46.95	0.027	100.70	10.13	70.97	54.49
26	65.00	2.00	1.00	30.00	5.00	40.00	0.1	0.9	39.07	0.142	223.59	16.35	59.36	58.99
27	65.00	3.00	0.10	6.00	8.00	20.00	0.55	0.8	24.62	0.153	33.03	2.12	42.44	41.66
28	65.00	4.00	0.40	14.00	11.00	0.00	0.4	0.7	21.93	0.276	81.82	5.35	50.34	61.50
29	90.00	1.00	0.70	30.00	5.00	20.00	0.55	0.7	59.81	0.040	196.45	16.60	46.61	56.21
30	90.00	2.00	1.00	22.00	2.00	0.00	0.4	0.8	45.83	0.098	77.06	8.80	50.37	71.64
31	90.00	3.00	0.10	14.00	11.00	60.00	0.25	0.9	32.04	0.323	143.31	6.32	69.77	64.62
32	90.00	4.00	0.40	6.00	8.00	40.00	0.1	1.0	27.50	0.272	49.69	2.56	24.95	53.96

**Table 4 materials-15-06790-t004:** Fitted equation.

Macroparameter	Fitted Equation	*R* ^2^
*E/GPa*	*E* = 0.461*E*_ƒ_ − 7.217*k*_ƒ_ + 0.477*σ_ƒ_* − 0.126*C*_ƒ_/*σ*_ƒ_ + 11.479	0.96
*ν*	*ν* = −0.001*E*_ƒ_ + 0.073*k*_ƒ_ + 0.002*σ*_ƒ_ + 0.011*C*_ƒ_/*σ*_ƒ_ − 0.062	0.86
*UCS* */MPa*	*UCS* = 0.482*E*_ƒ_ − 25.408*k*_ƒ_ + 6.245*σ*_ƒ_ + 11.084*C*_ƒ_/*σ*_ƒ_ − 16.497	0.83
*T* *S* */MPa*	*T**S* = 0.027*E*_ƒ_ − 1.509*k*_ƒ_ + 0.472*σ*_ƒ_ + 0.204*C*_ƒ_/*σ*_ƒ_ + 0.754	0.94
*φ*/°	*φ* = 5.207*k*_ƒ_ + 16.292*μ*_ƒ_ + 0.171*E*_ƒ_ + 0.201*φ_f_* + 17.754	0.68
*σ* _ci_	*σ*_ci_ = −55.493*R*_sd_ − 39.354$\theta_{b}\;$+ 0.708*σ*_ƒ_ + 104.25	0.664

**Table 5 materials-15-06790-t005:** Macroscopic mechanical parameters of Transjuane Sandstone [30,31].

Macro-Parameter	*E/*GPa	*ν*	*UCS**/*MPa	*T**S**/*MPa	*φ*/°	*σ* _ci_ _/_ *UCS*
Experiment value	12.5	0.3	40	2.8	41	0.42

**Table 6 materials-15-06790-t006:** Initial determination of microscopic parameters.

Micro-Parameter	*E*_ƒ_/GPa	*μ* _ƒ_	*k* _ƒ_	*C*_ƒ_/MPa	*σ*_ƒ_/MPa	*φ*_ƒ_/°	*R* _sd_	*θ* _b_
value	66.38	0.1	4.61	8.38	61.84	10	0.31	0.6

**Table 7 materials-15-06790-t007:** Initial determination of microscopic parameters.

Macro-Parameter	*E*/GPa	*ν*	*UCS**/*MPa	*T**S**/*MPa	*φ*/°	*σ* _ci_ _/_ *UCS*
Value	12.50	0.30	40.00	2.80	41.00	0.42
Initial determination	12.81	0.32	40.51	2.88	38.38	0.41
Error	2.5%	6.7%	1.3%	2.9%	6.4%	2.4%

**Table 8 materials-15-06790-t008:** The microscopic parameters are finally determined.

Micro-Parameter	*E*_ƒ_/GPa	*μ* _ƒ_	*k* _ƒ_	*C*_ƒ_/MPa	*σ*_ƒ_/MPa	*φ*_ƒ_/°	*R* _sd_	*θ* _b_
Value	64.23	0.12	4.59	8.31	61.71	10	0.31	0.6

**Table 9 materials-15-06790-t009:** Final soft rock calibration [29,30].

Macro-Parameter	*E*/GPa	*ν*	*UCS**/*MPa	*T**S**/*MPa	*φ*/°	*σ* _ci_ _/_ *UCS*
value	12.50	0.30	40.00	2.80	41.00	0.42
The final calibration	12.62	0.31	40.21	2.85	40.72	0.41
Error	0.96%	3.3%	0.5%	1.8%	0.7%	2.4%

## Data Availability

Data sharing is not applicable to this article.

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
