# Peer review of "A Quantitative Study of Micro and Macro Mechanical Parameters Based on the PFC3D Flat-Joint Model"

_materials, 2022, doi:10.3390/ma15196790_

Round 1

Reviewer 1 Report

-          The article has high similarity (about 19%) and should be reduced (Mostly similarity with JarosÅ‚aw JÄ™drysiak. "The Effect of the Material Periodic Structure on Free Vibrations of Thin Plates with Different Boundary Conditions", Materials, 2022; Chen Xu, Xiaoli Liu, Enzhi Wang, Sijing Wang. "Calibration of the Microparameters of Rock Specimens by Using Various Machine Learning Algorithms", International Journal of Geomechanics, 2021; "Proceedings of the 7th International Conference on Discrete Element Methods", Springer Science and Business Media LLC, 2017)

-          The results are not presented in the abstract section. The results should be presented briefly.

-          In Table 5, the authors have used the data of references 25 and 26, which belongs to soft rock, for validation. The compressive strength value of 40MPa or Ï•=41° cannot be acceptable for a soft rock. Usually, rocks with strength of less than 28MPa are considered soft.

-          Authors should describe what kind of rocks and with what kind of texture the laboratory data belong to.

Reviewer 2 Report

The manuscript ‘A quantitative study of micro and macro mechanical parameters based on the PFC3D flat-joint model’ is not appropriate for publication. However, maybe it can be improved (a lot) to be suitable for publication. These are my comments:

-          Round particles are not polygons

-          Introduction is too long with too many explanations. I would suggest to shorten introduction by reducing unnecessary information. On the other hand, introduction does not state what is new and scientific in this manuscript. Authors need to explain in Introduction what is the the main goal of the manuscript.

-          The reference list is also not appropriate since there are too many relevant references that are not included. These (and others) should be mentioned (they include methods such as DEM, FEM-DEM, lattice, embedded discontinuity, particles):

Lattice element models and their peculiarities, Archives of Computational Methods in Engineering 25 (3), 753-784, 2018

Y-Geo: new combined finite-discrete element numerical code for geomechanical applications, International Journal of Geomechanics 12 (6), 676-688, 2012

Confinement-shear lattice model for concrete damage in tension and compression: I. Theory, Journal of engineering mechanics 129 (12), 1439-1448,2003

Brittle and ductile failure of rocks: Embedded discontinuity approach for representing mode i and mode ii failure mechanisms, International Journal for Numerical Methods in Engineering 102 (8), 1507-1526 , 2015

Hybrid finite–discrete element modeling of geomaterials fracture and fragment muck-piling, International Journal of Geotechnical Engineering , Volume 9, 2015

Extended Rigid Body Spring Network method for the simulation of brittle rocks, Computers and Geotechnics 99, 31-41, 2018

-          The quality (resolution) of Figure 5 should be improved. The results in Figure 5 should be explained more precisely. The x and y axis and corresponding quantities should be marked

-          Figure 6 is too long. Not all of the results are necessary. Keep some that are most relevant and other can be explained through the text. The quality of Figure 6 is poor and should be improved. Other Figures are also of poor quality and should be improved.

-          In Figure 9, simulation and experiment are not even close when it comes to macroscopic responses. How do you comment on that?

-          You should elaborate how exactly the identification procedure (fitting) is performed. What are the measured responses that are relevant for identification? Is the problem ill-posed? Namely, can you find/obtain different combination of microscopic parameters that lead to the same macroscopic response? And how do you deal with that? These are important questions. You should reduce unnecessary sentences and elaborate precisely important questions about the general idea/approach in the paper. The bunch of results without proper order lead to unreadable manuscript.

-          You should also write some information about the model such as convergence properties, integration scheme etc.

-          English language should be corrected and improved. Repetitive sentences should be rewritten so that the quality of the manuscript is improved.

Round 2

Reviewer 1 Report

I received and have checked all author responses to my comments and suggestions. They have responded to all commutes and made all necessary corrections suggested in previous review on the manuscript. In my opinion, the paper can be accepted for publication now.
Yours faithfully

Reviewer 2 Report

The authors responded to my comments. I would further suggest to improve the quality of the figures and English language.